# Removal of Non-Specifically Bound Proteins Using Rayleigh Waves Generated on ST-Quartz Substrates

**DOI:** 10.3390/s22114096

**Published:** 2022-05-28

**Authors:** Mandek Richardson, Pradipta K. Das, Samuel Morrill, Kamlesh J. Suthar, Subramanian K. R. S. Sankaranarayanan, Venkat R. Bhethanabotla

**Affiliations:** 1Department of Chemical, Biological, and Materials Engineering, University of South Florida, Tampa, FL 33620, USA; mbrichar@mail.usf.edu (M.R.); pradipta@usf.edu (P.K.D.); samorri4@mail.usf.edu (S.M.); 2Center for Nanoscale Materials, Argonne National Laboratory, Argonne, IL 60439, USA; suthar@anl.gov (K.J.S.); ssankaranarayanan@anl.gov (S.K.R.S.S.)

**Keywords:** antibody, antigen, biosensing, non-specific binding, Rayleigh waves, surface acoustic wave (SAW)

## Abstract

Label-free biosensors are plagued by the issue of non-specific protein binding which negatively affects sensing parameters such as sensitivity, selectivity, and limit-of-detection. In the current work, we explore the possibility of using the Rayleigh waves in ST-Quartz devices to efficiently remove non-specifically bound proteins via acoustic streaming. A coupled-field finite element (FE) fluid structure interaction (FSI) model of a surface acoustic wave (SAW) device based on ST-Quartz substrate in contact with a liquid loading was first used to predict trends in forces related to SAW-induced acoustic streaming. Based on model predictions, it is found that the computed SAW body force is sufficient to overcome adhesive forces between particles and a surface while lift and drag forces prevent reattachment for a range of SAW frequencies. We further performed experiments to validate the model predictions and observe that the excitation of Rayleigh SAWs removed non-specifically bound (NSB) antigens and antibodies from sensing and non-sensing regions, while rinsing and blocking agents were ineffective. An amplified RF signal applied to the device input disrupted the specific interactions between antigens and their capture antibody as well. ST-quartz allows propagation of Rayleigh and leaky SH-SAW waves in orthogonal directions. Thus, the results reported here could allow integration of three important biosensor functions on a single chip, i.e., removal of non-specific binding, mixing, and sensing in the liquid phase.

## 1. Introduction

Non-specific binding (NSB) can occur when macromolecules attach to a sensor surface via weak interactive forces (i.e., Van der Waals, hydrophobic, and ionic) when it is in contact with a complex biological fluid (i.e., blood, urine, or serum). This is problematic for label-free sensing techniques because non-specific interactions cannot be distinguished from specific interactions. Current methods to reduce non-specific binding involve creating inert/resistant surfaces through modifications. Some examples are chemical attachment or physisorption of self-assembled monolayers (SAMs) with appropriate head groups and/or chain length [1,2], zwitterionic materials [3,4], blocking proteins (BSA) [5,6], or polymer films (i.e., poly(vinyl alcohol) (PVA), poly(vinylpyrrolidone) (PVP), and poly(ethylene glycol) (PEG)) [7,8,9,10]. Typical problems with these methods are increased setup time and costs due to additional steps and reagents, and incompatibility with common sensing materials.

The physical removal of NSB proteins has been accomplished using acoustic energy [11,12,13,14] and the physical processes involved have been elucidated using finite element simulations [13,15,16,17]. A study by Meyer et al. [11] showed that the shear waves generated by a quartz crystal microbalance (QCM) can remove weakly attached proteins by lowering the activation energy of desorption through the generation of mechanical stress at the interface. However, the relatively high power levels used (3.5 W) can generate heat that could result in loss of protein activity as demonstrated by previous studies [18,19,20]. In another study by Cular et al. [12,13] it was determined that NSB proteins could be removed via surface acoustic wave (SAW)-induced acoustic streaming [21,22,23]. Despite the success shown with this technique, the piezoelectric substrate used (128° YX LiNbO_3_) only supports Rayleigh waves, which have shown to be ineffective in liquid sensing applications. Therefore, another transduction mechanism would need to be employed for sensing, which would result in increased complexity.

In this work, the removal of NSB proteins is demonstrated theoretically and experimentally using Rayleigh SAWs generated on ST-quartz substrates. A coupled-field finite element (FE) fluid structure interaction (FSI) model of a SAW device in contact with a liquid loading was used to predict trends in forces related to SAW-induced acoustic streaming. Model predictions were utilized to compute the various interaction forces involved to determine if they are sufficient to remove non-specifically bound (NSB) proteins for a range of SAW frequencies. Experimentally, a micropattern of immobilized antibodies was applied to the delay path to segregate sensing and non-sensing areas. In successive steps, the removal of NSB antigens and an interfering protein from both areas, using SAWs, was studied. Lastly, an amplified RF signal was used to excite SAWs to examine if higher input power could disrupt antigen-antibody binding. Based on our results, a multifunctional “lab on a chip” device, with sensing and removal capabilities can be realized because quartz supports both shear-horizontal SAW (SH-SAW) used in biosensing, and Rayleigh SAW modes useful for NSB removal, on the same substrate.

## 2. Mechanism of NSB Protein Removal

The removal of non-specifically bound proteins depends on the relative magnitudes of the adhesive and the removal forces involved. The principal forces that non-specifically bind proteins immersed in a liquid to the SAW device surface are van der Waals and electrical double layer. For simplicity, proteins can be modeled as spherical particles with radius R. The van der Waals attraction force for a spherical particle near a flat surface is given by
(1)$$F_{vdW}\approx \frac{AR}{6z^{2}}$$
where A is the Hamaker constant for the non-retarded force and z is the distance where the force of adhesion is maximal. Typical values of A in a liquid environment are ~10–20 J. As z increases the van der Waals force becomes less significant. Typical values of z are in the range of 0.2–0.4 nm. Electrical double layer forces are typically associated with particles whose effective diameters are smaller than 5 μm [24]. In aqueous solutions, a surface contact potential is created between two different materials based on each material’s respective local energy state. To preserve charge neutrality, surface charges build up causing a double layer charge region. This results in electrostatic attraction between the materials. Since the magnitudes of these forces are similar, especially for sub-micron particles [25], it is usually sufficient to consider only the van der Waals force [26].

The forces responsible for the removal of particles are the direct SAW force and the lift and drag forces that result from the mean velocity field in the fluid. The magnitude of the direct force is given by [13].
(2)$$F_{SAW}\approx \sqrt{F_{x}^{2}+F_{z}^{2}}R^{2}$$

Here, *R* is the particle radius and the equations for the components $F_{x}$ and $F_{z}$ (force per unit area) can be found in [13]. The fluid circulation around particles results in inviscid lift forces, whose surface normal components act to remove the attached protein. The lift force is given by
(3)$$F_{L}\approx \rho_{f}{\left(u_{x}R\right)}^{2}$$
where *u_x_* and $\rho_{f}$ refer to the surface normal component of velocity and the fluid density, respectively. These forces can be estimated based on the Bernoulli’s equation by utilizing the pressure difference that exists between the bottom and top of the particle. The acoustic-streaming motion generated by SAWs leads to a drag force, caused by the interaction between the fluid mean flow and the particles. The drag force results from the boundary layer-generated acoustic streaming and is given by
(4)$$F_{ST}\approx \mu Ru_{z}$$

Here, *u_z_* and *μ* refer to the tangential component of velocity and the fluid viscosity, respectively.

Based on our previous coupled-field FE FSI simulations [13,15] and the forces described above, the mechanism for ultrasonic removal of NSB proteins is elucidated (Figure 1). The SAW direct force ($F_{SAW}$) simultaneously detaches NSB proteins from the SAW surface and moves them away from the region strongly influenced by adhesive forces (van der Waals). However, the SAW direct force decays rapidly with distance into the fluid—in the direction of wave propagation. To ensure removal occurs continuously and at distances where the direct force has diminished, a steady force is required. The hydrodynamic forces play a crucial role in this respect; the horizontal streaming-induced drag force ($F_{ST}$) helps to push the NSB proteins away from the fouled area whereas the vertical streaming force or the lift force ($F_{L}$) helps to prevent the re-adhesion of the proteins to the surface. As mentioned earlier, the van der Waals adhesion forces decrease rapidly with distance from the SAW surface. Thus, once the SAW force overcomes the adhesive forces and removes particles away from the surface, some distance z into the fluid, less force is required to prevent particle reattachment. At this point, lift ($F_{L}$) and drag forces ($F_{ST}$) are sufficient to keep the removal process ongoing. To remove NSB proteins for a particular system requires optimization of design parameters such as input power (or voltage) and SAW frequency.

## 3. Computational Details

A coupled-field FE FSI model, previously developed by Sankaranarayanan et al. [13] with modifications, was used to study acoustic steaming induced forces caused by SAW–fluid interaction. FE modeling of the fluid and solid domains requires a complex mathematical model function of the node shape factors and the equations of motion for each domain. Obtaining a solution to the FSI model involves sequentially solving the piezoelectric and fluid domains and transferring loads between each domain until convergence is achieved. The fluid flow fields are then computed to determine the various forces that act on particles in the system (i.e., non-specific proteins). The magnitudes of these computed forces are compared to the adhesive forces on particles to discern if they are sufficient in strength to cause removal of NSB proteins.

### 3.1. Model Setup

The model consisted of a SAW device based on ST-quartz substrate 400 μm wide × 800 μm long × 200 μm deep in contact with a 50-μm-thick fluid as shown in Figure 2. The input and output interdigital transducers (IDTs) each contained three finger pairs and were coupled by the voltage degree of freedom. The excitation of the piezoelectric solid was provided by applying an AC voltage at two different center frequencies (50 and 100 MHz) on the transmitter IDT fingers. The structure was simulated for a total of 100 nanoseconds (ns) with a time step of 1 ns.

### 3.2. Piezoelectric Domain

A system of four coupled wave equations for the electric potential and the three displacement components for a piezoelectric material are solved for the piezoelectric substrate or the solid domain [27]:(5)$$-\rho \frac{\partial^{2}u_{i}}{\partial t^{2}}+c_{ijkl}^{E}\frac{\partial^{2}u_{k}}{\partial x_{j}\partial x_{l}}+e_{kij}\frac{\partial^{2}\varphi }{\partial x_{k}\partial x_{j}}=0$$
(6)$$e_{ikl}\frac{\partial^{2}u_{k}}{\partial x_{i}\partial x_{l}}-\varepsilon_{ik}^{s}\frac{\partial^{2}\varphi }{\partial x_{i}\partial x_{k}}=0$$

Here, $c_{ijkl}^{E}$, $e_{kij}$, and $\varepsilon_{ik}^{s}$ are the elastic constant tensor for a constant electric field, the piezoelectric constant tensor, and the permittivity tensor, respectively, for constant strain; ρ is the density; $u_{i}$ are the mechanical displacements; φ is the electric potential; t is the time; and i, j, k, and l = 1,2,3. These coupled wave equations can be discretized and solved for generating displacement profiles and voltages at each element/node. The piezoelectric material displacements obtained from the above equations are applied to the solid model at each time step.

### 3.3. Fluid Domain

The Fluid domain was modeled using the Navier–Stokes equation; the arbitrary-Lagrangian–Eulerian approach was employed to handle the mesh distortions arising from the motion of the solid substrate.
(7)$$\rho_{f}\left(\frac{\partial {\vec{v}}_{f}}{\partial t}\right)+{\vec{v}}_{f}\cdot \nabla {\vec{v}}_{f}+\nabla P-2\eta \nabla \cdot \vec{D}=0$$
(8)$$\nabla \cdot {\vec{v}}_{f}=0$$

Here, ${\vec{v}}_{f}$, *P*, $\rho_{f}$, and *η* denote the fluid velocity, pressure, density, and viscosity, respectively. $\vec{D}$ is the rate of deformation tensor given by
(9)$$\vec{D}=\frac{1}{2}\left(\nabla {\vec{v}}_{f}+{\left[\nabla {\vec{v}}_{f}\right]}^{T}\right)$$

### 3.4. Coupling from Solid Domain to Fluid Domain

The fluid–solid coupling was established by maintaining stress and displacement continuity at the fluid–structure interface. The input velocity to the fluid domain is the velocity transferred from the piezoelectric domain at the interface. The boundary of fluid towards the atmosphere was kept as open with the following boundary condition:(10)$$u_{open}={\vec{n}}_{x}u_{x}+{\vec{n}}_{z}u_{z}$$
where ${\vec{n}}_{x}$ and ${\vec{n}}_{z}$ are the unit vectors along normal and tangential directions.

### 3.5. Coupling from Fluid Domain to Solid Domain

The structural deformations were solved using an elastic formulation and a nonlinear geometry formulation to allow large, time-varying deformations. The piezoelectric device is fixed to the bottom of the fluid channel and boundaries common between fluid and solid experience a load from the fluid, given by:(11)$$F_{T}=-\vec{n}\left(-P\;\vec{I}+\eta \left(\nabla {\vec{v}}_{f}+{\left[\nabla {\vec{v}}_{f}\right]}^{T}\right)\right)$$
where $\vec{n}$ is the normal vector to the boundary. This load term represents a summation of pressure and viscous forces acting on the substrate due to velocity.

## 4. Computational Results and Discussion

### 4.1. Streaming Induced Removal of Non-Specifically Bound Proteins in ST-Quartz

To determine if Rayleigh SAWs generated on ST-Quartz are able to remove NSB proteins, a coupled-field FE FSI model was used to predict the various SAW-induced acoustic streaming forces. The main mechanisms responsible for overcoming adhesive forces (van der Waals and electrical double layer) are the direct SAW force (*F_SAW_*), lift (*F_L_*), and drag forces (*F_ST_*) that result from the mean velocity field in the fluid [15]. An AC voltage (1 V) of varying frequency (1, 10, 50, and 100 MHz) was applied to the transmitting IDT, and the streaming velocities were obtained along the fluid film thickness at distances of 0.5λ and 2.5λ (λ = SAW wavelength) along the delay path (Figure 3).

Our simulated results show that the streaming velocity increases with the SAW frequency (Figure 3). The simulated streaming velocities are in the µm/s range for the frequencies studied, which are consistent with experimental values found in the literature for streaming flows induced by SAW devices [28,29,30]. As the frequency of the surface-acoustic-wave device increases, it would lead to an increase in the irrotational velocity. This in turn results in an increase in the acoustic-streaming velocity, which is proportional to the square of the irrotational velocity. Note, however, that the acoustic-streaming velocity is also proportional to the square of the amplitude of SAW displacement. The simulated SAW amplitudes tend to increase with decreasing device frequency. This has also been observed in the experimental studies of Sano et al. [31] and Shiokawa et al. [32]. Therefore, the increase in SAW amplitude somewhat offsets the corresponding decrease in the device frequency. As a result, the simulated SAW streaming velocities shown in Figure 3 exhibit a near-linear variation.

For a given applied voltage and device frequency, the removal of non-specifically bound proteins depends on the relative magnitudes of the adhesive and the removal forces involved. The principal adhesive forces are the van der Waals and the electrical double layer involved in the binding of specific and non-specific proteins to the SAW surface. The magnitudes of these forces are similar, especially for sub-micron particles and hence for the sake of comparison, it is usually sufficient to consider only the van der Waals force which is provided in Table 1 [25,33].

The forces involved in NSB protein removal were estimated based on the streaming velocities given in Figure 3 (see Section 2 for calculation details). The SAW direct force ($F_{SAW}$) as well as lift ($F_{L}$) and drag forces ($F_{ST}$) generated for several SAW frequencies are given in Table 1. Our results show that for each frequency studied, the direct SAW force ($F_{SAW}$) is greater than adhesive forces ($F_{vdW}$) by several orders of magnitude. Comparison of the computed direct SAW force ($F_{SAW}$) at different frequencies shows that this force increases as frequency decreases. Therefore, particles would be ejected into the fluid at a greater distance at 10 MHz, in comparison to that at 50 and 100 MHz. Analysis of the other forces (lift ($F_{L}$) and drag ($F_{ST}$)) presented in Table 1 reveals that their magnitude is insufficient to overcome adhesive forces at the surface by themselves. However, it was mentioned in Section 2 that van der Waals force ($F_{vdW}$) decreases with increasing z into the fluid. Thus, initial removal would occur via the direct SAW force ($F_{SAW}$). Subsequently, particles would be removed from the surface, some distance z, to a region where the van der Waals force ($F_{vdW}$) has diminished to a point where lift ($F_{L}$) and drag forces ($F_{ST}$) are strong enough to prevent particle reattachment. As a result, particle removal is maintained. Please note here that specific bindings involve long-range forces such as ionic or hydrophobic bonds which show much stronger interaction compared to the short-range Van der Waals force [34]. Hence our proposed mechanism can effectively be used to eliminate non-specific interferences in real sensors.

### 4.2. Comparison of the Removal Efficiency amongst the Various SAW Substrates

In this section, we briefly compare the efficiencies of removal amongst various commonly used substrates for SAW sensing and microfluidic applications. We had previously demonstrated the removal of non-specific proteins via acoustic streaming induced by Rayleigh wave in a LiNbO_3_ substrate [13]. It is worth noting that most microfluidic applications that utilize SAWs are based on LiNbO_3_ [35]. However, with respect to sensing, one of the drawbacks is that the substrate (128° YX LiNbO_3_) utilized does not support an SH-SAW. Thus, integration of removal with sensing requires a separate sensor platform.

One of the viable alternatives is langasite, which allows propagation of both SH-SAW and Rayleigh SAW in orthogonal directions. Singh et al. [17] simulated orthogonally placed transducers on langasite and demonstrated two different wave modes to propagate (i.e., pure SH-SAW and Rayleigh wave). This would allow sensing and removal to be accomplished on the same substrate via SH-SAW and Rayleigh SAW, respectively. However, an experimental device has not been realized to date. Furthermore, langasite is expensive and extremely fragile so that fabrication yield would be an issue.

An alternative is to use ST-Quartz because it is more robust mechanically and less expensive. In addition, it supports both Rayleigh waves and a surface skimming bulk wave (SSBW) that has SH particle polarization [36,37,38]. Several investigators over the years have used the SSBW wave mode for biosensing in liquids [39,40,41]. However, the Rayleigh wave in ST Quartz compared to LiNbO_3_ and Langasite has a weaker surface normal component owing to its weaker electromechanical coupling coefficient. Therefore, a relatively higher power is required for the Rayleigh waves generated on ST-Quartz to generate a force sufficient to overcome NSB.

We, therefore, compare the primary removal and adhesive forces induced by Rayleigh waves for the three common SAW substrates, i.e., 128° YX LiNbO_3_, langasite (0,22,0), and ST-Quartz (shown in Table 2). The various removal forces, namely the SAW body force, lift force, and drag force were computed for the three substrates using the simulated fluid velocities as outlined by Sankaranarayanan et al. [13] and compared with the nonspecific adhesive force. The IDT finger periodicity was 40 microns for all the substrates. An input voltage of 10 V was applied over 3 IDT finger pairs. The fluid viscosity is 1 cP. The calculated forces are based on the streaming velocity field generated at the IDT region.

It can be seen that the primary adhesive force binding the non-specific proteins is the van der Waals force which is ~2 × 10^−8^ N. It is clear that the SAW body force is the primary removal force that overcomes the adhesive force and moves away from the surface until the nonspecific adhesive force decreases significantly. Although the magnitude of the SAW body force in ST-Quartz is two orders of magnitude lower than 128° YX LiNbO_3_ and an order of magnitude lower than langasite, it is still two orders of magnitude larger than the adhesive force binding the non-specific proteins to the device surface. This suggests that initial removal of proteins from the substrate should definitely be feasible.

The fluid-induced lift and drag forces then come into play away from the surface to prevent NSB protein reattachment to the surface and facilitate its removal from the fluid stream, respectively. A quick comparison suggests that the induced hydrodynamic forces in ST-Quartz are higher compared to those in 128° YX LiNbO_3_ and langasite. This is not surprising since the coupling between the Rayleigh wave and the fluid domain is not as strong as in 128° YX LiNbO_3_ and langasite and hence the wave dissipation is not as strong. The particle advection induced by the hydrodynamic forces prevents the reattachment of the removed proteins back to the device surface. Given the practical advantages of utilizing ST-Quartz over a LiNbO_3_ and langasite substrate, our analysis of the various adhesive and removal forces suggests that complete removal of non-specific proteins via streaming forces should definitely be possible.

## 5. Materials and Methods

### 5.1. SAW Device Fabrication

To confirm the simulation predictions, a Rayleigh SAW device was fabricated on ST-Quartz wafer 0.5 mm thick and 100 mm in diameter using standard lithography, metal evaporation, and lift-off. Each IDT consisted of a Ti/Au film 20/80 nm thick and contained 60 finger pairs. The aperture height = 50λ and the distance between IDTs is 200λ. The operating wavelength, λ, is 40 μm. This corresponds to an operating frequency of 78.95 MHz.

### 5.2. Micropatterning the SAW Delay Path

A square micropattern was applied to the entire delay path to differentiate between sensing and non-sensing areas (Figure 4a) [11,12]. Before applying the micropattern, SAW chips were rinsed with acetone, methanol, isopropanol, DI water, and dried with an N_2_ stream. Next, a positive tone Microposit S1813 photoresist (Shipley) was spun onto the wafer at 3000 rpm for 40 s followed by soft baking on a hotplate at 115 °C for 60 s. After cooling for ~5 min, the resist was exposed to broadband UV for 3 s, at 25 mW/cm^2^. The substrate was then immersed in AZ 300 MIF Developer (Clariant) for 45 s to define square windows to the substrate surface. This was followed by rinsing with DI water, drying with a stream of N_2_, and plasma descumming (Harrick Plasma) for 5 min at the lowest setting. Lastly, the wafer was diced into individual die to be utilized in subsequent experiments.

### 5.3. Surface Modification and Attachment of Capture Antibody

An organosilane film ((3-glcidoxypropyl) dimethylethoxysilane (3-GPDMS) (Sigma Aldrich)), was used to functionalize exposed areas on the SAW device surface to immobilize the capture antibody (rabbit IgG (Sigma Aldrich, St. Louis, MI, USA)) (Figure 4b). First, the SAW chip was immersed in 3-GPDMS (1 *v*/*v*%) in toluene for 1 h. Next, the devices were thoroughly rinsed with toluene, dried with a stream of N_2_, and baked in an oven at 125 °C for 1 h to complete the hydrolysis reaction. Upon cooling, the photoresist was removed by immersing devices in acetone. Afterward, rabbit IgG (10 μL, 100 µg/mL) in PBS (pH 7.4) was applied to the center of the delay path for 45 min. This was followed by rinsing 3 times with PBS (pH 7.4) and drying with a stream of N_2_. To block the non-sensing regions BSA (10 μL, 1 mg/mL) in PBS (pH 7.4) was applied to the surface for 45 min (Figure 4c), followed by rinsing three times with PBS (pH 7.4) and drying with N_2_.

### 5.4. Determination of Specific and Non-Specific Binding

The determination of specific and non-specific binding in both sensing and non-sensing regions was accomplished by using fluorescently labeled antibodies. Thus, the extent that binding occurred was quantified from fluorescent intensity measurements. The specific protein (Alexa Fluor 488 goat anti-rabbit IgG (Molecular Probes)) was immobilized by covalent attachment to the silane layer (Figure 4c). Alexa Fluor 488 goat anti-rabbit IgG (10 μL, 100 µg/mL) in PBS (pH 7.4), was applied to the SAW delay path center and left for 30 min. After incubation, the devices were rinsed 3 times with PBS and imaged using a fluorescence microscope (Leica). After imaging, we applied a control (Alexa Fluor 488 donkey anti-mouse IgG (Molecular Probes)) to ascertain the degree to which non-specific binding occurs. Alexa Fluor 488 donkey anti-mouse IgG (10 μL, 100 µg/mL) in PBS (pH 7.4) was applied to the delay path center and left for 30 min. Similarly, after incubation, the device was rinsed 3 times with PBS (pH 7.4) and fluorescence images were taken.

### 5.5. Fluorescence Imaging

Fluorescence images were taken with a Leica DMI 4000B fluorescence microscope (Leica Microsystems, Wetzlar, Germany). Prior to imaging, the SAW device was inverted and mounted on a glass slide covered with a thin PBS film. A 20X objective lens along with a Leica DFC340-FX monochrome digital camera (Leica Microsystems) were used for all image capturing. The microscope had the following settings: Γ = 1.0, gain = 1.3, and exposure time = 3.6 s. Schematic of the experimental setup is shown in Figure 5. For Alexa Fluor 488 we used the I3 filter cube with an excitation filter with a bandpass of 450–490 nm. Image analysis was completed using ImageJ software. For quantitative analysis, we defined signal as the fluorescent intensity of the sensing regions and background as the fluorescent intensity of the non-sensing regions.

### 5.6. Protein Removal with Acoustic Energy

An RF signal from an Agilent 8753 ES network analyzer (Agilent) was used to excite the SAW device at its center frequency in order to remove non-specifically bound proteins. To account for frequency shifts due to liquid loading or non-specific protein detachment, a 1 MHz bandwidth around the center frequency was used. The initial input power supplied to the device was 0 dBm. The input power multiplied by the excitation time (10 min) defines the dose. To see if the sensor surface could be regenerated, an amplified signal provided by a TIA-1000-1R8 RF broadband amplifier (Mini-Circuits) was inserted into the input signal path. The amplifier provided a 40.6 dB signal gain over the frequency range used to interrogate the SAW device. After the application of RF power, the liquid was removed from the SAW device. Fluorescence images of the SAW delay path were taken and compared to images taken after a standard rinsing step.

## 6. Experimental Results and Discussion

In this section, we discuss the results from experimental trials involving a SAW device based on ST-Quartz that generates a Rayleigh wave mode in which its ability to remove non-specifically bound proteins from its surface was determined. Three scenarios were examined where protein removal with sound waves is desirable: (1) non-specific attachment of antigens in sensing and non-sensing areas; (2) non-specific attachment of an interfering species in sensing and non-sensing areas; and (3) specific attachment of antigens to its corresponding antibody in sensing areas.

### 6.1. Removal of Non-Specifically Bound Antigen from Sensing and Non-Sensing Regions

During biosensing, the antigen of interest can weakly bind to other antigens on the surface and to non-sensing areas producing an inflated response. Antigens (Alexa 488 goat anti-rabbit IgG) non-specifically bound to sensing and non-sensing areas were assessed by creating a micropattern, containing the capture antibody (rabbit IgG) in defined areas. To compare the amount of non-specific interaction, two images were taken. First, a fluorescent image was taken after application of the antigen and performing a standard rinsing step with PBS (Figure 6a). Then, PBS (10 μL) was placed on to the delay path and RF power was applied to the input of the SAW device for 10 min. After rinsing we obtained a second image of the SAW delay path (Figure 6c). Figure 6a shows that the square pattern we tried to obtain is occluded by a fluorescent signal of higher intensity occurring randomly over the entire surface which is due the agglomeration of antigen molecules. In Figure 6c, taken after the application of acoustic energy, the fluorescent signal in random locations shown in Figure 6a was reduced significantly and the desired square pattern is clearly visible. The results suggest that antigens bind to other antigen molecules, which is evident by the high intensity signal caused by their agglomeration (Figure 6a). In addition, antigens will also bind to sensing and non-sensing regions indiscriminately seen by the randomness of the fluorescent signal shown in Figure 6a. After comparing the results in Figure 6a,c, we conclude that the antigen-binding is relatively weak and can be characterized as non-specific. Furthermore, the results show that acoustic energy has the ability to remove weakly bound antigens where rinsing alone has failed. Not only has non-specifically bound material been removed, but antigens covalently immobilized to the patterned region stay intact (Figure 6c). Thus, removal can be accomplished without disrupting specific binding between an antigen and its capture antibody. To quantitatively determine the amount of antigen removed in both sensing and non-sensing regions, we averaged the fluorescent intensity for both areas before and after application of acoustic energy. The intensity values were normalized to values obtained before excitation of SAWs. Results given in Figure 6d show that the fluorescent intensity decreased by 83% in the sensing areas and 86% in non-sensing areas.

### 6.2. Removal of an Interfering Protein from Sensing and Non-Sensing Regions

In many biosensing applications, an interfering species will be present in the sample to be analyzed and these species can non-specifically bind to the sensor surface. Using a similar approach to the one in Section 6.1, sensing and non-sensing regions are differentiated with a micropattern. To determine to what extent an interfering species would non-specifically bind to sensing and non-sensing areas we applied Alexa Fluor 488 donkey anti-mouse IgG to the SAW delay path. To establish if a SAW can remove proteins that weakly bind to the surface, we compared a fluorescence image after a standard rinsing step (Figure 7a) to an image taken after applying acoustic energy (Figure 7c). In this setup, antigens (goat anti-rabbit IgG) specifically bound in sensing regions were already present from the previous experiment. The results after application of the interfering species and rinsing with PBS are given in Figure 7a. Similar to the results presented in Figure 7a, we observed a fluorescent signal randomly distributed over the entire surface. A comparison of Figure 7a–c shows that the use of acoustic energy substantially reduces the fluorescent signal outside of the patterned region where specific binding occurs. Due to the reduction in the fluorescent signal over the entire area and maintenance of the fluorescent signal in areas where specific binding takes place, we attribute most of the randomly located signals to non-specific binding of the interfering species. Just like in Section 6.1, these results show that SAWs removed non-specifically bound proteins, and rinsing alone was not sufficient. To measure the reduction in fluorescent intensity due to the removal of non-specifically bound proteins, we plotted the intensities normalized to the value obtained before the application of acoustic energy for both sensing and non-sensing regions (Figure 7d). The fluorescent intensity was reduced by 35% in sensing regions and 94% in non-sensing areas. These results further indicate that non-specific binding of the interfering species occurs in both areas, and they can be effectively removed with acoustic energy.

### 6.3. Renewal of Sensor Surface

Regeneration of a sensor surface for its subsequent reuse can reduce costs in certain instances. For immunosensing applications, this would require rupturing the bond between the antigen and capture antibody. To determine if the antigen (goat anti-rabbit IgG) can be detached from its capture antibody (rabbit IgG) using acoustic energy, we increased the input signal to the SAW device by 35 while PBS (10 μL) covered the delay path. To establish if the higher input power was effective in reversing antigen–antibody binding, we compared fluorescence images before and after generating SAWs. Figure 8a is an image of the delay path after the conclusion of the experiment in Section 6.2. At this point, antigens are attached to their capture antibody in defined regions. Results after the application of an amplified RF signal to the SAW input are shown in Figure 8b,c. The image in Figure 8b is taken directly at the center of the SAW delay path. We observed that after the application of acoustic energy the fluorescent intensity decreased. To quantify the relative decrease, we plotted the average intensity in the sensing areas normalized to the intensity before the application of acoustic energy (Figure 8d). Our results show that the average intensity in the sensing areas decreased by 71.6%. This indicates that antigen–antibody binding is disrupted. Because the SAW’s energy is dissipated as it propagates into the fluid, removal efficiency is greater near the excitation source. This was tested by taking an image of the delay path close to the input IDT (Figure 8c). Examination of Figure 8c shows that the fluorescent signal is barely visible. This illustrates that all antigens are possibly removed from their capture antibodies. However, it is conceivable that both capture antibodies and antigens have been removed from the organosilane film.

## 7. Conclusions

A combined theoretical and experimental study was performed to evaluate whether acoustically induced removal of non-specifically bound proteins is possible on ST-Quartz substrate which is a commonly used platform for biosensing. A coupled field FE FSI model was first used to predict SAW-induced acoustic streaming forces. The various forces involved in NSB protein removal were computed based on predictions from our model and compared to adhesive forces binding these non-specific proteins. For a range of SAW frequencies, computations show that the SAW body force is sufficient to overcome adhesive forces between particles and the device surface. In addition, our model suggests that lift and drag forces permit continuous removal to occur by not allowing particle re-attachment.

Based on the predicted mechanism of removal, we carried out experimental investigations to successfully remove non-specifically bound antigens and interfering proteins using Rayleigh SAWs on ST-Quartz. The micropattern consisted of immobilized antibodies separated sensing areas from non-sensing areas, while the use of fluorescent tags allowed qualitative and quantitative verification. A comparison of fluorescent intensity values before and after acoustic excitation shows intensity is reduced by 83% in sensing areas and 86% in non-sensing areas, confirming the removal of NSB antigens using SAWs. Similarly, the removal of weakly bound interfering proteins was proved by a 35% and 94% reduction in fluorescent intensity in sensing and non-sensing areas, respectively. Lastly, an amplified RF signal was applied to the SAW input to disrupt specific binding between antigens and capture antibodies. The corresponding fluorescence images showed that binding was disrupted in the middle of the delay path (71.6% reduction in fluorescent intensity) but was more effective near the input IDT (~100% reduction in fluorescent intensity).

The results presented in this work reveal that a standard rinsing technique and application of a blocking agent (BSA) are not adequate to prevent biofouling. Furthermore, the non-specific binding of interfering species and antigens produce an inflated response that can have negative implications in clinical applications. The demonstrated removal of NSB proteins using Rayleigh SAWs on ST-Quartz opens up the possibility of an integrated SAW biosensor that can simultaneously eliminate biofouling and sense biomarkers.

## Figures and Tables

**Figure 1 sensors-22-04096-f001:**
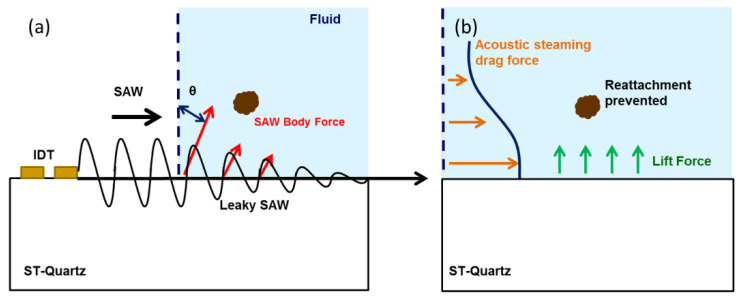
Coupled-field FE FSI predicted mechanism of ultrasonic removal of particles weakly bound to a surface (i.e., NSB proteins). The solid line in (**a**) represents the amplitude of the SAW. The dashed line presents the surface normal and θ represents the Rayleigh angle. The black arrow represents the propagation direction of the SAW. Simulations suggest that the interaction of the Rayleigh wave with the fluid medium results in a wave mode conversion into leaky SAW. The SAW is attenuated in the liquid medium as a result of this mode conversion. This leaky SAW propagates along the boundary between the piezoelectric and the fluid loading and excites longitudinal waves into the fluid (red arrows) at a Rayleigh angle θ as seen in (**a**). The longitudinal wave propagation leads to the generation of hydrodynamic forces (drag and lift) in the liquid medium. SAW direct forces result in initial particle detachment whereas hydrodynamic forces (i.e., drag (F_ST_) and lift (F_L_) forces are shown by orange and green arrows, respectively) prevent their reattachment as shown in (**b**).

**Figure 2 sensors-22-04096-f002:**
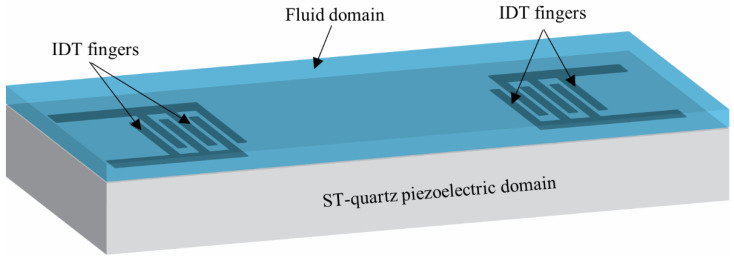
Schematic of the computational domain considered for the study. It consists of a piezoelectric domain (ST-quartz) of 400 μm × 800 μm × 200 μm. The fluid domain of 50 μm thick is considered at top of the piezoelectric substrate. Three pairs of input and output IDTs are considered and AC voltages of 50 and 100 MHz were imposed on the input IDTs.

**Figure 3 sensors-22-04096-f003:**
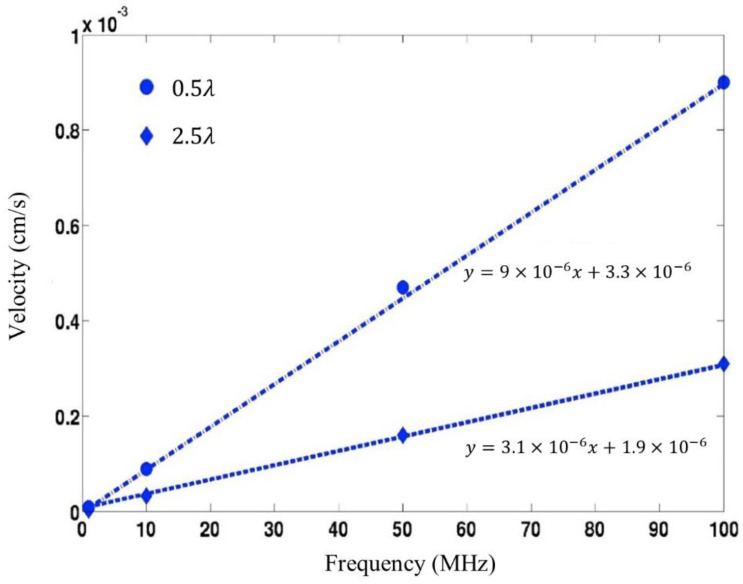
Streaming velocity along the delay path with respect to device frequency. Variation along two different regions in the delay path (i.e., 0.5 λ and 2.5 λ) is shown.

**Figure 4 sensors-22-04096-f004:**
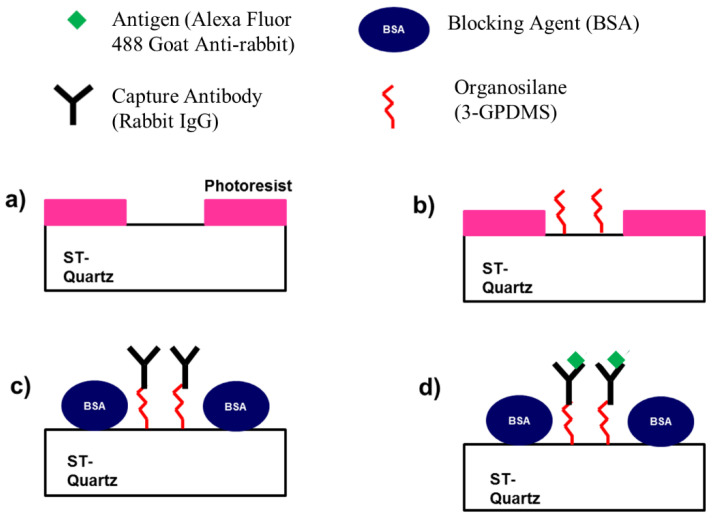
Processing steps to create patterned areas where specific binding occurs. (**a**) Photoresist is patterned to expose areas on ST-Quartz substrate. (**b**) Exposed areas are functionalized with organosilane film (3-GPDMS). (**c**) Capture antibody (rabbit IgG) is covalently immobilized and BSA is applied to block non-sensing areas. (**d**) Fluorescently tagged antigen (Alexa Fluor 488 goat anti-rabbit IgG) is specifically attached to capture antibody (rabbit IgG).

**Figure 5 sensors-22-04096-f005:**
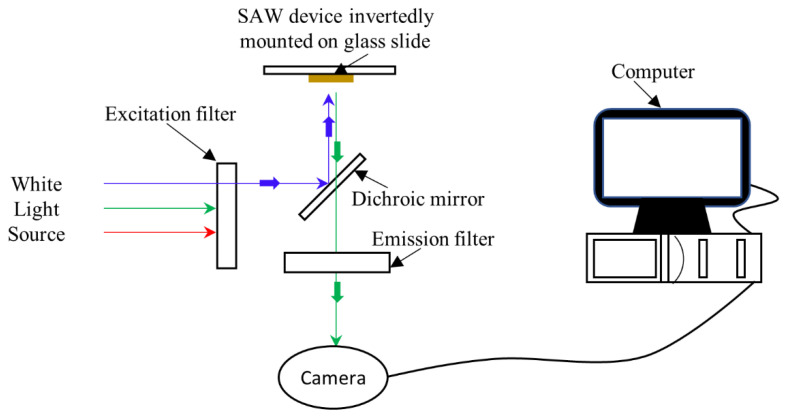
Schematic of the experimental setup showing inverted fluorescence microscope, connected to a computer for image capture and processing. The SAW sample is inverted and mounted on the glass slide to measure the fluorescence.

**Figure 6 sensors-22-04096-f006:**
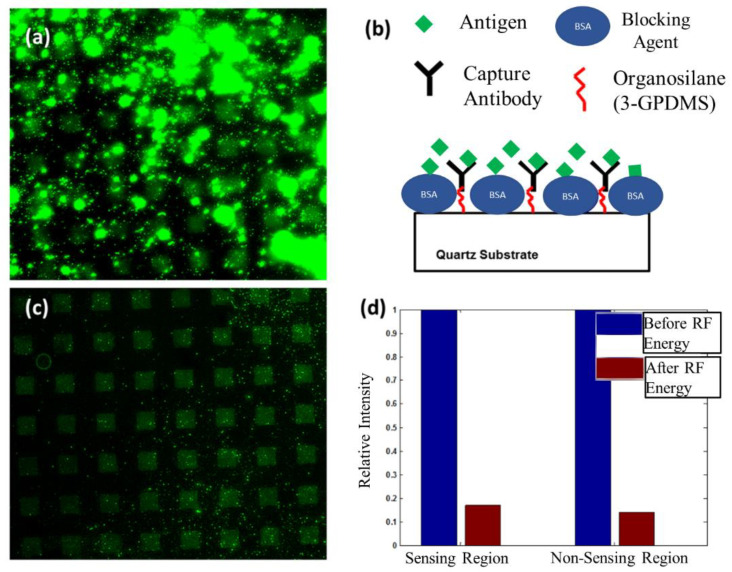
Results indicating the removal of non-specifically bound antigens (goat anti-rabbit IgG) from the SAW delay path. (**a**) Fluorescent intensity image shows binding of Alexa 488 tagged antigen (goat anti-rabbit IgG) to sensing and non-sensing regions after incubation and rinsing. (**b**) Illustration of the experimental setup. (**c**) Fluorescent intensity image taken at the center of the delay path after applying acoustic energy to remove non-specifically bound antigen (goat anti-rabbit IgG) from sensing and non-sensing regions. (**d**) Plot of relative fluorescent intensity for sensing and non-sensing regions with respect to values obtained before application of acoustic energy.

**Figure 7 sensors-22-04096-f007:**
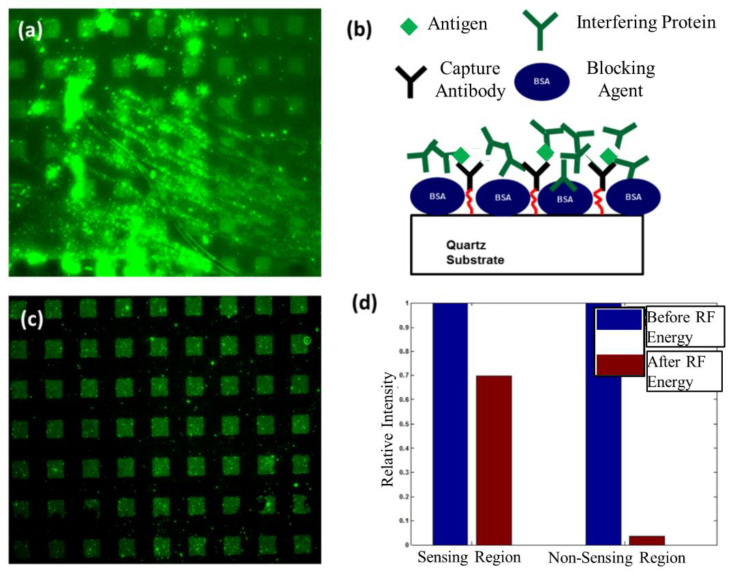
Results indicating the removal of an interfering protein (donkey anti-mouse IgG) from the SAW delay path. (**a**) Image of fluorescent intensity shows Alexa 488 tagged interfering protein (donkey anti-mouse IgG) binds to sensing and non-sensing regions after incubation and rinsing. (**b**) Illustration of the experimental setup. (**c**) Image of fluorescent intensity taken at the center of the delay path after applying acoustic energy. (**d**) Plot of relative fluorescent intensity for sensing and non-sensing regions with respect to values obtained before application of RF energy.

**Figure 8 sensors-22-04096-f008:**
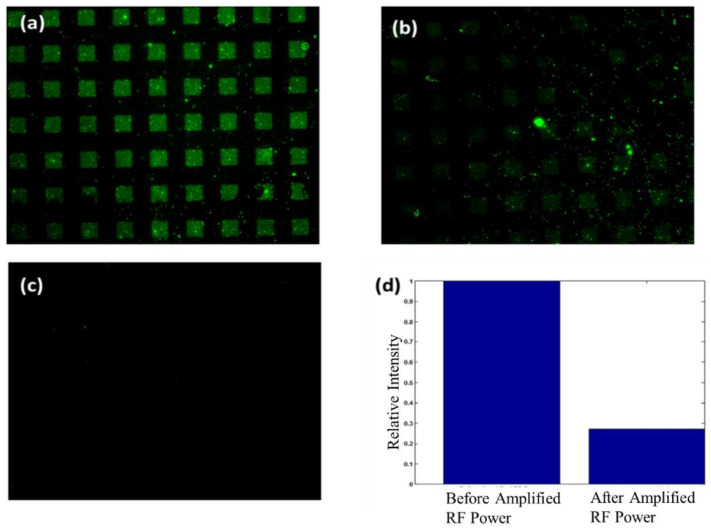
Results after applying an amplified RF signal to the SAW input. (**a**) Image of fluorescent intensity shows Alexa 488 tagged antigen (goat anti-rabbit IgG) bound to sensing regions after non-specifically bound proteins were removed with SAWs as outlined in Section 5.2. (**b**) Image of fluorescent intensity at the middle of the delay path after applying an amplified RF signal. (**c**) Image of fluorescent intensity near the input IDT after applying an amplified RF signal. (**d**) Plot of relative fluorescent intensity for sensing regions with respect to values obtained before application of an amplified RF signal.

**Table 1 sensors-22-04096-t001:** Forces (in Newtons) as a function of device frequency for ST-Quartz. The particle radius is taken to be 1 μm and the applied voltage is 25 V.

Force	10 MHz	50 MHz	100 MHz
*F_vdW_*	2 × 10^−8^	2 × 10^−8^	2 × 10^−8^
*F_SAW_*	2 × 10^−4^	8 × 10^−5^	1 × 10^−5^
*F_L_*	4 × 10^−10^	2 × 10^−9^	4 × 10^−8^
*F_ST_*	4 × 10^−10^	8 × 10^−10^	2 × 10^−9^

**Table 2 sensors-22-04096-t002:** Forces (in Newton) as a function of applied voltages for a ~100 MHz SAW device. The particle radius is taken to be 1 micron. The fluid viscosity is 1 cP. The calculated forces are based on the streaming velocity field generated at the IDT region. An applied voltage of 10 V is chosen simply for the purpose of comparing the SAW removal mechanism.

Force	128° YX LiNbO_3_	Langasite (0,22,0)	ST-Quartz
*F_vdW_*	2 × 10^−8^	2 × 10^−8^	2 × 10^−8^
*F_SAW_*	4 × 10^−4^	3.3 × 10^−5^	5 × 10^−6^
*F_L_*	2 × 10^−14^	1 × 10^−14^	2 × 10^−10^
*F_ST_*	1 × 10^−11^	4 × 10^−11^	1 × 10^−10^

## Data Availability

The data that support the findings of this study are available within the article.

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
