# Peer review of "Removal of Non-Specifically Bound Proteins Using Rayleigh Waves Generated on ST-Quartz Substrates"

_sensors, 2022, doi:10.3390/s22114096_

Round 1

Reviewer 1 Report

Richardson et al describe an interesting method for removing nonspecifically bound material from a sensor surface, using Rayleigh waves. Given the well-known and widely studied problem of nonspecific binding on surfaces used for label-free biosensing, this is a potentially significant advance. Both the computational work and experimental results are consistent with the authors’ hypothesis. However, in my view several revisions to the manuscript and some additional experimental work will be required before this can be suitable for publication in Sensors.

1) Given that the “removal of non-specifically bound proteins depends on the relative magnitudes of the adhesive and the removal forces involved”, what’s the minimum difference between “adhesive” and “removal” forces that will result in either adhesion or removal? This is information that would be highly useful to readers designing sensors, as it relates to the question of how weak a “specific” dissociation constant the system can tolerate will still allowing removal of nonspecifically bound material.

2) No schematic is provided of the experimental measurement system, and no diagram is given showing the position of the piezoelectric device. Schematics should also be provided both for the system analyzed computationally.

3) The potential use of this technique to regenerate sensors is particularly intriguing, and the authors’ results in Figure 6 suggest this may indeed be possible. It is puzzling, however, that the authors did not then attempt to re-use the sensor, since (as stated in lines 482-483) “it is conceivable that both capture antibodies and antigens have been removed from the organosilane film”. This is an essential (and simple) experiment to do, because otherwise the regeneration protocol is of no use.

Additional minor comments

Figures 3, 4, and 5: The authors show antigen binding between the two “arms” of the IgG. Although this reviewer recognizes that this is only a cartoon, it is still important to provide representations that are as scientifically accurate as possible. IgG molecules have two antigen binding sites, one on the end of each “arm”.

Line 338: “fluorescence microscope”, not “fluorescent microscope” (presumably the microscope itself is not fluorescent). Likewise lines 344 and 345 should be “Fluorescence imaging” and “Fluorescence images”, respectively. Similar comments through the remainder of the manuscript.

Author Response

Richardson et al describe an interesting method for removing nonspecifically bound material from a sensor surface, using Rayleigh waves. Given the well-known and widely studied problem of nonspecific binding on surfaces used for label-free biosensing, this is a potentially significant advance. Both the computational work and experimental results are consistent with the authors’ hypothesis. However, in my view several revisions to the manuscript and some additional experimental work will be required before this can be suitable for publication in Sensors.

1) Given that the “removal of non-specifically bound proteins depends on the relative magnitudes of the adhesive and the removal forces involved”, what’s the minimum difference between “adhesive” and “removal” forces that will result in either adhesion or removal? This is information that would be highly useful to readers designing sensors, as it relates to the question of how weak a “specific” dissociation constant the system can tolerate will still allowing removal of nonspecifically bound material.

Response: We thank the reviewer for the comment. Van der Waals and electric double layer forces are responsible for the adhesiveness where they are comparable with each other. Hence, we only calculated Van der Waals interactions to present the adhesive force. On the other hand, SAW force is the removal force acting on the non-specifically attach proteins to detach them from the surface, and ideally, the removal force must be greater than the Van der Waals force so that the non-specific adhesion can be avoided.

On the other hand, specific bindings such as antigen-antibody bindings (often known as epitope-paratope bindings) are much stronger than non-specific adhesion and usually occur within a small portion of the molecule (area ~ 0.4 to 8 nm2). These bindings happen when the epitope and the paratope come very close to each other (~ few nanometers) and long-range forces such as hydrophobic or ionic bonds cause the attractive force to specifically bind them.

Table 1 of reference [34] shows the relative strength of different interactions.

To illustrate the difference between specific and non-specific adhesion, we have added the following sentences and a reference to our revised manuscript

----------------------------------------------------------------------------------------

Please note here that specific bindings involve long-range forces such as ionic or hydrophobic bonds which show much stronger interaction compared to the short-range Van der Waals force [34]. Hence our proposed mechanism can effectively be used to eliminate non-specific interferences in real sensors without damaging specific bindings.

Reference

[34 ] Reverberi, R.; Reverberi, L., Factors affecting the antigen-antibody reaction. Blood Transfus 2007, 5, (4), 227-240.

----------------------------------------------------------------------------------------

2) No schematic is provided of the experimental measurement system, and no diagram is given showing the position of the piezoelectric device. Schematics should also be provided both for the system analyzed computationally.

Response: We have provided schematics of computational domain and the fluorescence measurement experiments in the revised manuscript (see Figure 2 and Figure 5).

3) The potential use of this technique to regenerate sensors is particularly intriguing, and the authors’ results in Figure 6 suggest this may indeed be possible. It is puzzling, however, that the authors did not then attempt to re-use the sensor, since (as stated in lines 482-483) “it is conceivable that both capture antibodies and antigens have been removed from the organosilane film”. This is an essential (and simple) experiment to do, because otherwise the regeneration protocol is of no use.

Response: We thank the reviewer for the comment. The regeneration experiments show that the average intensity of the sensing region (where Alexa 488 tagged antigen was bound to capture antibody) has been decreased by 71.6% after applying SAW of 40.6 dBm. In fact, using SAW based non-specific proteins removal technique, the fluorescence intensity in the sensing region was reduced by 35% only whereas the intensity in the non-specific binding region dropped by 94%. This gives us a contrast ratio of ~10.8:1 between the sensing and non-sensing region intensity. After applying 40.6 dBm SAW for the regeneration, the intensity in the sensing region decreased down to 18.5% of the original intensity. Based on this calculation, regeneration seems feasible. Although, exact experiments can provide us with much more detailed information, we could not perform additional experiments due to the time constraints.

Please note here, non-specific binding removal is the most significant barrier in point of care (POC) sensing from a drop of blood or plasma. Achieving that with SAW devices, as shown in the present work, has a huge potential and these SAW chips are very cheap (one dollar or less). Hence, even if the use of regeneration techniques is a good idea and appears feasible, however, without regeneration also, we have major advances in terms of non-specific binding removal efficiencies. Regenerating the chip is a minor advance and in health-care settings, it will not be trusted even if we were successful in regenerating and they would rather throw away the chip for a dollar or less, much like advancing reusable gloves.

Additional minor comments

Figures 3, 4, and 5: The authors show antigen binding between the two “arms” of the IgG. Although this reviewer recognizes that this is only a cartoon, it is still important to provide representations that are as scientifically accurate as possible. IgG molecules have two antigen binding sites, one on the end of each “arm”.

Response: We have corrected those Figures to show the binding on one end of the antibody arms.   

Line 338: “fluorescence microscope”, not “fluorescent microscope” (presumably the microscope itself is not fluorescent). Likewise lines 344 and 345 should be “Fluorescence imaging” and “Fluorescence images”, respectively. Similar comments through the remainder of the manuscript.

Response: We have corrected those terms.

Reviewer 2 Report

The manuscript „Removal of Non-Specifically Bound Proteins Using Rayleigh Waves Generated on an ST-Quartz Substrate” 

For the rreaders' convenience, the authors are kindly requested to address the following aspects:

  • carefully check that are defined all variables that are present in each equation;
  • the text in the figures is not sufficiently visible (the same font and size should be used for all figures);
  • highlight the novelty of the present study in comparison to other published results on the subject.

Author Response

The manuscript „Removal of Non-Specifically Bound Proteins Using Rayleigh Waves Generated on an ST-Quartz Substrate” 

For the readers' convenience, the authors are kindly requested to address the following aspects:

  • carefully check that are defined all variables that are present in each equation;

Response: We have checked carefully, and all variables are defined in each equation.

  • the text in the figures is not sufficiently visible (the same font and size should be used for all figures);

Response: We have increased the visibility of each and every figure in the revised manuscript.

  • highlight the novelty of the present study in comparison to other published results on the subject.

Response: We thank the reviewer for the suggestion. The novelty of the work lies in the experimental and theoretical results that show the possibility of ST-quartz to be used as a platform to eliminate non-specifically attached protein and improve biosensor performances. Section 4.2 compares the present results with the relevant published results.

Round 2

Reviewer 1 Report

The authors have done a good job of addressing my comments from the previous review. I believe this is suitable for publication in Sensors.